# Vitamin D Status Is Not Associated with Cognitive or Motor Function in Pre-School Ugandan Children

**DOI:** 10.3390/nu12061662

**Published:** 2020-06-03

**Authors:** Agnes M. Mutua, Margaret Nampijja, Alison M. Elliott, John M. Pettifor, Thomas N. Williams, Amina Abubakar, Emily L. Webb, Sarah H. Atkinson

**Affiliations:** 1Kenya Medical Research Institute (KEMRI), Centre for Geographic Medicine Research-Coast, KEMRI Wellcome Trust Research Programme, P.O. BOX 230-80108 Kilifi, Kenya; Tom.n.williams@gmail.com (T.N.W.); AAbubakar@kemri-wellcome.org (A.A.); 2Maternal and Child Wellbeing (MCW) Unit, African Population and Health Research Center, P.O. Box 10787-00100 Nairobi, Kenya; maggie.nampijja@gmail.com; 3Medical Research Council/Uganda Virus Research Institute and London School of Hygiene and Tropical Medicine Uganda Research Unit, P.O. Box 49, Entebbe, Uganda; Alison.Elliott@lshtm.ac.uk; 4Department of Clinical Research, London School of Hygiene and Tropical Medicine, London WC1E 7HT, UK; 5South African Medical Research Council and Wits Developmental Pathways for Health Research Unit, Department of Paediatrics, University of the Witwatersrand, 26 Chris Hani Road, Soweto 6201, Johannesburg, South Africa; John.Pettifor@wits.ac.za; 6Department of Medicine, Imperial College of Science Technology and Medicine, St Mary’s Hospital, London W2 1NY, UK; 7Centre for Tropical Medicine and Global Health, Nuffield Department of Medicine, University of Oxford, Oxford OX3 7FZ, UK; 8Department of Public Health, School of Human and Health Sciences, Pwani University, P.O. BOX 195-80108 Kilifi, Kenya; 9Department of Psychiatry, University of Oxford, Oxford OX3 7JX, UK; 10Institute for Human Development, Aga Khan University, 2nd Parklands Avenue, P.O. BOX 30270-00100 Nairobi, Kenya; 11MRC Tropical Epidemiology Group, Department of Infectious Disease Epidemiology, London School of Hygiene and Tropical Medicine, London WC1E 7HT, UK; Emily.Webb@lshtm.ac.uk; 12Department of Paediatrics, University of Oxford, Oxford OX3 9DU, UK

**Keywords:** vitamin D, cognitive function, motor function, children, Africa

## Abstract

Vitamin D deficiency is common worldwide and young children are among the most affected groups. Animal studies suggest a key role for vitamin D in brain development. However, studies investigating the effects of vitamin D on neurobehavioural outcomes in children are inconclusive and evidence is limited in sub-Saharan Africa. We evaluated the effect of vitamin D status on cognitive and motor outcomes using prospective data from the Entebbe Mother and Baby Study birth cohort. We analysed data from 302 Ugandan children with 25-hydroxyvitamin D (25(OH)D) measurements below five years and developmental measures at five years of age. We used multivariable linear regression, adjusted for potential confounders, to estimate the effect of 25(OH)D on cognitive and motor outcomes. Of 302 children, eight (2.7%) had 25(OH)D levels <50 nmol/L, 105 (35.8%) had levels 50–75 nmol/L and 189 (62.6%) had levels >75 nmol/L. There was no evidence that earlier vitamin D status was associated with cognitive and motor outcomes in five-year-old Ugandan children. This study adds to the sparse literature and highlights the need for further longitudinal studies on vitamin D and neurobehavioural outcomes in children living in sub-Saharan Africa.

## 1. Introduction

The first five years in a child’s life are a period of rapid brain development [1]. About 81 million children aged three to four years have poor cognitive and socioemotional development in low and middle-income countries worldwide and 29.4 million of these children live in sub-Saharan Africa [2]. Therefore, mitigation of recognized risk factors for impaired development, including infections, inadequately stimulating home environments and malnutrition, is necessary [3]. Evidence indicates that micronutrient deficiencies may also be an important risk factor for impaired neurobehavioural outcomes [4]. Vitamin D deficiency and/or insufficiency is prevalent among populations worldwide [5,6] and children are among the most at-risk groups for sub-optimal 25-hydroxyvitamin D (25(OH)D) levels [7,8]. A recent systematic review and meta-analysis of vitamin D deficiency in Africa reported a pooled prevalence of vitamin D deficiency (25(OH)D levels <50 nmol/L) of 49.1% among newborns and 23.0% among children below 18 years of age [9].

Vitamin D may influence neurobehavioural outcomes through its role in brain development. The presence of vitamin D receptors and metabolites in cerebrospinal fluid and parts of the brain including the hippocampus and the cortex may indicate a key role for vitamin D in brain development [10]. Vitamin D deficiency is also associated with structural and functional alterations in the developing brains of rodents, which are associated with impaired behaviour, learning, motor and cognitive processes [11,12,13]. Furthermore, vitamin D appears to plays a role in neuroprotection through its anti-inflammatory properties in the brain [14]. There are few epidemiological studies of the effect of vitamin D status on neurobehavioural outcomes in children and their findings are inconsistent [15]. One small randomized controlled trial, without a placebo arm, reported a beneficial effect of supplementation with lower doses of vitamin D compared to higher doses [16] while another found no beneficial effect of vitamin D supplementation on cognitive development in extremely preterm infants compared to no supplementation [17].

Only two studies have investigated the relationship between the vitamin D status of children and their neurobehavioural outcomes in Africa, and these were in school-aged children. A prospective study of 254 Ugandan children reported associations between 25(OH)D levels and socio-emotional adjustment outcomes among school-children who were stratified according to perinatal HIV exposure/infection and in utero/peripartum antiretroviral exposure [18], while a cross-sectional study of 45 Egyptian school-children reported a positive association between higher 25(OH)D levels and improved cognition and school performance compared to lower levels [19]. Our study aimed to evaluate the association between vitamin D status and cognitive and motor outcomes in pre-school Ugandan children using data from the Entebbe Mother and Baby Study (EMaBS) prospective birth cohort. We analysed data from 302 community-based children with 25(OH)D measurements below five years of age and cognitive and motor development assessed at five years of age.

## 2. Materials and Methods

### 2.1. Ethical Approval

Ethical approval was given by the Medical Research Council (MRC)/Uganda Virus Research Institute Research Ethics Committee (reference numbers: GC/127 and GC/127/15/04/35), Uganda National Council for Science and Technology and the London School of Hygiene and Tropical Medicine Ethics Committee (reference numbers: 790 and 17261). Parents or guardians provided informed consent in writing, or with a thumbprint if not literate with a signature from a literate witness.

### 2.2. Participants and Study Design

The Entebbe Mother and Baby Study (EMaBS) prospective birth cohort was initially designed as a double-blind randomized controlled trial of the effects of anthelmintic treatment during pregnancy and early childhood on immunological and disease outcomes in childhood (ISRCTN32849447). Details of the study are described elsewhere [20]. Briefly, between April 2003 and November 2005, 2507 pregnant women were recruited into the study during their first antenatal care clinic visit at Entebbe hospital and randomized to receive either albendazole or placebo and praziquantel or placebo in a 2 × 2 factorial design. Their children formed the EMaBS birth cohort that was followed up from birth and seen during scheduled annual visits or when they were sick. The children were randomized to receive albendazole or placebo from the age of 15 months to five years and children with helminth infection detected at annual visits were treated according to clinical guidelines. Data on sex, birthweight, gestational age and anthropometric measurements were recorded and blood samples were taken at birth and during annual visits. Children were brought to the study clinic for diagnosis and treatment when they were unwell. Information on maternal age, socioeconomic status, location, education, parity and medical conditions was collected at enrolment. Household socioeconomic status was derived as a composite of building materials of the home, number of rooms and items owned. A total asset score was calculated and categorized into 6 ordered categorical levels with 1 indicating lowest and 6 indicating highest household socioeconomic status.

### 2.3. Sample Size

In the larger EMaBS study, the cohort size of 2500 was determined based on the primary immunological objectives. For the present sub-study, assuming a prevalence of 36.6% of 25(OH)D levels ≤75 nmol/L among Ugandan children [21] and a standard deviation of two in cognitive and motor scores [22], our sample size of 302 has 80% power at α = 0.05 to detect a difference of 0.7 when comparing mean cognitive/motor scores between the 25(OH)D levels ≤75 nmol/L and >75 nmol/L groups.

### 2.4. Laboratory Procedures and Definitions

Stored blood samples were analysed based on availability with most samples being from two years of age. The biomarkers assayed were plasma 25(OH)D (Chemiluminescent Microparticle Immunoassay (CMI), Abbott Architect), ferritin (CMI, Abbott Architect), haemoglobin (Medonic CA 530) and C-reactive protein (CRP, MULTIGENT CRP Vario assay, Abbott Architect) [23]. For this study, we classified 25(OH)D levels into three categories; levels <50 nmol/L, 50–75 nmol/L and >75 nmol/L, according to the Endocrine Society Practice Guidelines [24]. We defined iron deficiency as plasma ferritin <12 µg/L and inflammation as CRP levels >5 mg/L. Anaemia was defined as haemoglobin levels <11 g/dL, after subtracting 0.2 g/dL from all haemoglobin values to adjust for altitude (1000 m above sea level). Malaria parasitaemia was measured using Giemsa-stained thick and thin blood smears and a malaria episode was defined as parasitaemia and temperature >37.5 °C. All children were tested annually for asymptomatic malaria and the study included longitudinal active surveillance for malaria and other infections during fortnightly home visits and quarterly clinic visits. Children of HIV-positive mothers were tested for HIV at six weeks and 18 months using a filter-paper blood spot for DNA-PCR, a measure for HIV-antibodies and viral RNA.

### 2.5. Cognitive and Motor Assessments

At age five years, cognitive and motor tests were performed at the EMaBS clinic by trained nurses and doctors in 60–90-min sessions including breaks. Of the 13 measures used to assess cognitive and motor abilities at five years, four (counting span, running memory, shapes task and tower of London) were excluded in this analysis as they were administered to a small sub-set of children (*n* < 113). We combined the remaining nine individual measures using principal components analysis to generate scores on three components of child development; verbal and non-verbal intelligence quotient (IQ), executive function (EF) and motor function. We excluded two measures (tap one tap twice task and sentence repetition) that did not load well on the IQ, EF or motor function components. We excluded one recorded observation in the coinbox test and another in the balancing on one leg test as they were beyond the maximum attainable scores for the respective measures. Details of the translation and validation of all of the neurobehavioural measures in the Ugandan setting are described elsewhere [25]. In a small pilot study, involving children not in the current study, the measures had good internal consistency (Cronbach’s alpha = 0.65 − 0.82) and test-retest reliability (correlation coefficient (r) = 0.45 − 0.88), except the Wisconsin card sort test (r = 0.22) which was administered to only 19 children in the pilot study [25]. The measures used for the current study are briefly described in Table 1.

### 2.6. Statistical Analysis

All analyses were conducted using STATA version 15.0 (StataCorp, College Station, TX 77845, United States of America). We assessed the distribution of 25(OH)D levels and developmental scores using histograms and scatterplots. We examined the characteristics of the participants using descriptive statistics including proportions and means with standard deviations and used ANOVA to compare mean scores between children with 25(OH)D levels <50 nmol/L, 50–75 nmol/L and >75 nmol/L. We compared baseline characteristics of children included in these analyses (*n* = 302) and those who were not included (*n* = 2043). Of the 2043 children, 594 were lost to follow-up, 116 died, 312 did not attend their fifth annual visit, 540 did not have cognitive and motor outcome data and 481 children did not have data on 25(OH)D levels prior to five years of age.

We used univariable and multivariable linear regression models to estimate the effect of vitamin D in early childhood on cognitive and motor outcomes at five years. We identified potential confounders from literature including age at 25(OH)D measurement, sex, iron status and anaemia (haemoglobin levels <11 g/dL), inflammation (CRP levels >5 mg/L), height-for-age Z-scores, randomized treatment of children with albendazole, socioeconomic status, maternal education and asymptomatic malaria parasitaemia. We empirically tested associations of the potential confounders identified from literature in addition to weight-for-age and weight-for-height Z-scores, location, maternal age, parity and randomized maternal treatment with albendazole and praziquantel with 25(OH)D levels and cognitive and motor outcomes using multivariable linear regression models. Then, using a forward modelling approach, we fitted three separate linear regression models for each outcome starting with separate crude models with 25(OH)D levels as the exposure variable (included as a continuous variable, per 10 nmol/L increase) for each of the three outcomes. Then we generated minimally adjusted models with age at vitamin D measurement, sex, iron and anaemia status included as potential confounders. Finally, we added other potential confounders one by one to the minimally adjusted models observing the change in regression coefficients until final multivariable models for each of the three outcomes were generated. We re-ran these final models, fitting vitamin D as a categorical variable (25(OH)D levels ≤75 nmol/L compared to levels >75 nmol/L). We were unable to further investigate associations with 25(OH)D levels <50 nmol/L using multivariable analysis since few children (*n* = 8) had 25(OH)D levels <50 nmol/L, so children with 25(OH)D levels <50 nmol/L were grouped with children that had levels 50–75 nmol/L. We also compared children with 25(OH)D levels 50–75 nmol/L with children that had levels >75 nmol/L.

Due to missing observations in haemoglobin (16 missing values) and ferritin levels (14 missing values), we assessed whether excluding these variables in final models impacted results. We examined multicollinearity using the variance inflation factor (VIF) which measures change in the regression coefficient if predictor variables are correlated, with a VIF > 10 suggesting severe multicollinearity. None of the variables had a VIF > 2 in any model although clinical malaria episodes and asymptomatic malaria parasitaemia appeared to be moderately correlated and we therefore, only adjusted for asymptomatic malaria parasitaemia.

## 3. Results

### 3.1. Description of Study Participants

A total of 302 children, 48.3% boys and 51.7% girls were included in this study. None of the included children or their mothers received vitamin D supplementation. Characteristics of the study participants are presented in Table 2. Most children (68.2%) had 25(OH)D levels measured at the second annual visit (mean age = 2.3 years, standard deviation (SD) = 0.7) and mean age at cognitive and motor assessment was 5.0 (SD = 0.02) years. Mean (SD) and median (interquartile range) 25(OH)D levels were 83.2 (22.2) nmol/L and 81.0 (67.9 to 95.5) nmol/L, respectively. Eight (2.7%) children had 25(OH)D levels <50 nmol/L, 105 (34.8%) had levels 50–75 nmol/L and 189 (62.6%) had levels >75 nmol/L.

In this population, stunting at age 5 years was more common (34.4%) than underweight (11.1%) or wasting (4.7%). Malaria was common, with 55.3% children having had at least one episode of clinical malaria while 11.3% had asymptomatic malaria parasitaemia at an annual visit between birth and five years. Over a third of the children were anaemic, while 27.1% were iron deficient and 23.1% had inflammation at the time of 25(OH)D measurement. The majority of participants resided in urban (39.1%) or peri-urban (26.2%) areas (Table 2). Of 302 children, four (1.3%) were HIV positive at 18 months and of 300 children with available data, four (1.7%) were delivered preterm (<37 gestational weeks). Baseline characteristics were comparable between participants included in the analyses and those not included, with the exception that stunting was more common while asymptomatic malaria parasitaemia was less common among participants than those not included. Participants were more likely to have higher socioeconomic status while mothers of participants were older and had higher parity compared to children who were not included (Appendix A).

### 3.2. Associations Between 25(OH)D and Participant Characteristics

In the multivariable analyses we observed evidence of a decreasing trend of 25(OH)D levels with increasing child age and asymptomatic malaria was associated with high 25(OH)D levels. Inflammation was associated with higher mean 25(OH)D levels and 25(OH)D levels >75 nmol/L were more common among those with inflammation than for those with lower CRP levels (Table 2).

### 3.3. Cognitive and Motor Outcomes at Five Years and Associations with Participant Characteristics

The IQ, EF and motor function scores were normally distributed. Mean scores for the individual tests that were used to assess cognitive and motor outcomes are presented in Appendix A. Girls had higher EF and motor scores compared to boys after adjusting for potential confounders, but there was no difference by sex for IQ scores (Table 3). Children who were randomly allocated placebo in the original EMaBS trial had lower motor scores compared to those who received albendazole, but little difference was observed in IQ and EF scores (Table 3). Stunting, low socioeconomic status and low levels of maternal education were associated with lower IQ scores in univariable analyses but not after adjusting for potential confounders. No associations were observed with other participant characteristics in univariable analyses (Appendix A).

### 3.4. Associations between 25(OH)D and Cognitive and Motor Outcomes

Table 4 shows the univariable and multivariable linear regression results for the effect of 25(OH)D levels on cognitive and motor outcomes. We observed little association between 25(OH)D levels and IQ, EF or motor function either in crude analysis or after adjusting for potential confounders. Considering 25(OH)D as a categorical variable, we found no evidence of differences in the mean scores among children with 25(OH)D levels <50 nmol/L, 50–75 nmol/L and >75 nmol/L for IQ (*p* = 0.89), EF (*p* = 0.44) or motor function (*p* = 0.44) outcomes. We found little evidence of association between IQ, EF or motor function and 25(OH)D levels ≤75 nmol/L compared to levels >75 nmol/L or for levels 50–75 nmol/L compared to levels >75 nmol/L after adjusting for potential confounders (Table 4).

## 4. Discussion

We evaluated the association between 25(OH)D levels measured in young Ugandan children below five years and their cognitive and motor outcomes at five years of age. In this study population, we found that few children had 25(OH)D levels <50 nmol/L (2.7%), and the majority of children (62.6%) had 25(OH)D levels >75 nmol/L. We found no evidence of association between vitamin D status and IQ, EF or motor function. This finding may be explained by the small proportion of children with 25(OH)D levels <50 nmol/L limiting the ability to detect an effect, especially if 25(OH)D levels <50 nmol/L, rather than 25(OH)D levels ≤75 nmol/L, drives impaired neurobehavioural outcomes [33]. Girls had higher EF and motor scores compared to boys but there was no difference by sex for IQ scores. Children who were randomly allocated placebo in the original trial had marginally lower motor scores compared to those who received albendazole, but we did not observe differences in IQ and EF scores.

Our findings of no association between vitamin D status and cognitive or motor function, agree with nine previous observational studies [34,35,36,37,38,39,40,41,42]. On the contrary, a small dose-response randomized controlled trial (*n* = 55) of vitamin D supplementation in Canadian infants reported that a lower dose of 400 international units (IU) of vitamin D was more beneficial for gross motor development than higher doses of 800 IU or 1200 IU [16]. This finding may indicate a threshold effect for vitamin D but may be limited by the small sample size and lack of a control group that did not receive vitamin D supplements. Additionally, a cross-sectional study of one-year-old Iranian children (*n* = 186) reported a positive association between high 25(OH)D levels and improved motor function [43]. To our knowledge, ours is the first prospective study to investigate the relationship between vitamin D status and cognitive and motor development in pre-school children in sub-Saharan Africa. To date, only three observational studies have evaluated the effect of child or maternal vitamin D status on neurobehavioural outcomes in African children. One prospective study (*n* = 254) observed an association between increasing 25(OH)D levels and improved socioemotional adjustment in Ugandan school-aged children who were HIV negative and those perinatally exposed to HIV but uninfected and unexposed to early antiretroviral therapy (ART) [18]. They also observed an association between increasing 25(OH)D levels and poor socioemotional adjustment in HIV-infected children and in HIV-uninfected children exposed to early ART, which may be attributed to dysregulation of 25(OH)D metabolism by ART. Another small study (*n* = 45) reported that high 25(OH)D levels were positively associated with improved cognition in Egyptian school-aged children [19]. Additionally, an observational study of 202 mother-child pairs in Seychelles reported no association between maternal 25(OH)D levels during pregnancy and neurobehavioural outcomes at five years [44].

Contrary to our findings, animal and in vitro studies provide consistent evidence for the adverse effects of vitamin D deficiency on neurodevelopment and possible mechanisms for this effect. The presence of vitamin D receptors and metabolism enzymes in parts of the brain involved in cognition suggests that vitamin D may be important for cognition [45]. Similarly, the presence of vitamin D receptors in skeletal muscles suggests that vitamin D is important for muscle development and subsequent motor function. Experimental evidence shows reduced type II muscle fibre diameter in mice whose skeletal muscle vitamin D receptors are knocked out [46]. Furthermore, vitamin D deficiency is implicated in alterations in neurotransmitter synthesis and synaptic plasticity which may affect learning processes, cognition [11,47] and motor processes [13,48]. However, the application of these findings to humans may be limited as experimental studies are able to generate conditions of adverse vitamin D deficiency to an extent unlikely to be observed in humans. Additionally, it is unclear when 25(OH)D levels might influence neurobehavioural outcomes. Some observational studies in mother-child pairs suggest that levels of 25(OH)D influence neurobehavioural outcomes exclusively during the prenatal period [49,50,51]. However, other similar studies have reported little association between maternal 25(OH)D levels and neurobehavioural outcomes in children [52,53].

We observed a decreasing trend of 25(OH)D levels with increasing child age consistent with previous evidence [54]. Younger children may acquire 25(OH)D from their mothers through placental transfer of 25(OH)D in utero or breastfeeding during infancy. Asymptomatic malaria parasitaemia was associated with higher 25(OH)D levels. Evidence on the relationship between malaria and 25(OH)D levels is limited and inconclusive [55,56]. High 25(OH)D levels were associated with increased inflammation contrary to other studies that have shown possible anti-inflammatory properties of vitamin D [57]. We found that two other factors influenced neurobehavioural outcomes. Girls performed better in the EF and motor tests compared to boys, consistent with previous studies [58,59]. There is no clear evidence of the causes for the observed differences though they may be attributed to differences in environmental and socio-cultural factors between boys and girls [60]. It is also possible that the differences were a developmental pattern that changed after the pre-school age [61]. Furthermore, randomized treatment of children with albendazole in the EMaBS trial was marginally associated with higher motor scores compared to placebo treatment. This may be a chance finding since in the larger EMaBS birth cohort no associations were observed between either child or maternal treatment with albendazole and cognitive or motor scores [62,63].

Strengths of our study include the extensive battery of tools used to measure cognitive and motor outcomes that were specifically validated and adapted to the Ugandan population [25]. Moreover, prospectively collected data on multiple covariates were available allowing adjustment for many potential confounders. Furthermore, we analysed 25(OH)D both as continuous and categorical variables which was important considering the lack of global consensus on vitamin D status cut-offs. Several limitations must be considered when interpreting our results. We included about 13% of participants in the original EMaBS birth cohort in our analyses as 25(OH)D levels and cognitive and motor outcomes were assessed in smaller subsets of participants and over a quarter of the original participants were lost to follow up. However, our study was adequately powered to detect the reported effect sizes, and selection bias was unlikely as most baseline characteristics were comparable between children who were included in these analyses and those who were not included. We were unable to assess the association between 25(OH)D levels <50 nmol/L and cognitive and motor outcomes as only 2.7% children had 25(OH)D levels <50 nmol/L in this study population. Additionally, plasma 25(OH)D levels were measured at one time-point prior to developmental assessment and may not reflect the vitamin D status of the children throughout early childhood or at the time of cognitive and motor assessment, which could lead to underestimation of the association between 25(OH)D and cognitive or motor outcomes. Further, the possibility of other unmeasured variables that may affect 25(OH)D levels cannot be ruled out. These include habits that might influence exposure to sunlight, for example, outdoor physical activity and clothing and dietary vitamin D intake. However, diet may not be an important risk factor since vitamin D fortification and supplementation is not routine in Uganda. There may have also been other unmeasured factors that might have influenced cognitive and motor outcomes, such as pre-schooling, inadequately stimulating home environment or maternal IQ. Additionally, other variables recoded at enrolment such as location and socioeconomic status of some participants may have changed over the study period which may possibly result in residual confounding for the associations between 25(OH)D and cognitive and motor outcomes.

## 5. Conclusions

In conclusion, our findings add to the evidence from similar studies that report no association between vitamin D status and cognitive and motor outcomes in children. This study adds to the very limited literature on the effect of vitamin D status on neurobehavioural outcomes in children from sub-Saharan Africa. Further longitudinal studies in populations with 25(OH)D levels <50 nmol/L and well-conducted randomized controlled trials of vitamin D supplementation would be valuable in clarifying the causal effect of vitamin D status in early childhood for cognitive and motor development.

## Figures and Tables

**Table 1 nutrients-12-01662-t001:** Description of the cognitive and motor measures used in the study.

PCA Components	Name of Test	Domain	Description of Measure	Absolute Scores (Min, Max)
Verbal and non-verbal IQ	Block design	Non-verbal IQ	The measure is adapted from the British Ability Scales-third edition [26]. The child is asked to copy and construct items with wooden blocks following a demonstration by the assessor.	0, 16
	Picture vocabulary scale	Verbal IQ	The measure is adapted from the Kilifi Vocabulary Test [27]. The child is asked to point out and identify items from 24 black and white picture items familiar to them.	0, 24
Executive function	Verbal fluency	Working memory	The measure is adapted from the Developmental NEuroPSYchological Assessment [28]. The child is asked to name items including foods and animals as fast as possible in a minute.	*
	Picture search	Selective attention	The measure is adapted from the Sky Search in Tests of Everyday Attention for Children [29]. The child is presented with three A3 sheets each with a target picture on top and about 100 others at the bottom including copies of the target picture. The child is asked to locate as many copies of the target pictures as possible within 10 s.	**
	Wisconsin card sort test	Cognitive flexibility	The measure is adapted from Berg’s card sort test [30]. The child is given four playing cards of different suits and a pack of 12 cards and asked to sort the cards by number (block 1) and suit (block 2).	0, 12
Motor function	Coin box	Fine motor function	The measure is adapted from the Kilifi Developmental Inventory [31]. The child is asked to slot coins through a small opening on a coinbox within 20 s in two trials.	0, 20 †
	Balancing on one leg	Gross motor function	The measure is adapted from the Movement Assessment Battery for Children [32]. It entails timed attempts (two per leg) of balancing on one leg for one minute.	0, 60 †

PCA, principal components analysis; IQ, intelligence quotient; Min, minimum score; Max, maximum score. *One point is awarded for each correct name and a total score is calculated from the total correct names in a minute. **A total score is calculated from the number of target pictures identified within 10 s. † An average score is calculated after timed attempts of the tests. Block design and picture vocabulary tests loaded heavily on verbal and non-verbal IQ while verbal fluency, picture search and Wisconsin card sort tests loaded heavily on executive function. Coinbox and balancing on one leg tests loaded heavily on motor function. The resulting scores were centred on zero with higher scores representing better and lower scores worse development.

**Table 2 nutrients-12-01662-t002:** Baseline characteristics stratified by 25(OH)D levels of 302 Ugandan children included in the analysis and multivariable linear regression results for associations between 25(OH)D and participant characteristics.

Characteristics	All Participants	25(OH)D Levels ≤75 nmol/L) (*n* = 113)	25(OH)D Levels >75 nmol/L (*n* = 189)	25(OH)D Levels (Per 10 nmol/L) (*n* = 295) β (95% CI) ††	*p* Value
Age at 25(OH)D measurement in years, (*n*, %)					
1	10 (3.3)	3 (30.0)	7 (70.0)	0.91 (−0.79, 2.62)	0.02 *
2	206 (68.2)	75 (36.4)	131 (63.6)	Reference	
3	63 (20.9)	24 (38.1)	39 (61.9)	−0.49 (−1.19, 0.19)	
4	23 (7.6)	11 (47.8)	12 (52.2)	−1.31 (−2.72, 0.11)	
Sex (*n*, %)					
Male	146 (48.3)	56 (38.4)	90 (61.6)	Reference	
Female	156 (51.7)	57 (36.5)	99 (63.5)	−0.29 (−0.82, 0.24)	0.29
Height-for-age Z-scores (mv = 8)					
Normal (>2 SD)	193 (65.7)	79 (40.9)	114 (59.1)	Reference	
Stunted (<2 SD)	101 (34.4)	32 (31.7)	69 (68.3)	0.21 (−0.36, 0.78)	0.47
Weight-for-age Z-scores (mv = 5)					
Normal (>2 SD)	264 (88.9)	103 (39.0)	161 (61.0)	Reference	
Underweight (<2 SD)	33 (11.1)	8 (24.2)	25 (75.8)	0.33 (−0.57,1.23)	0.47
Weight-for-height Z-scores (mv = 5)					
Normal (>2 SD)	283 (95.3)	105 (37.1)	178 (62.9)	Reference	
Wasted (<2 SD)	14 (4.7)	6 (42.9)	8 (57.1)	−0.50 (−1.84, 0.84)	0.46
Helminthic infections between birth and 5 years (*n*, %)					
Negative	248 (82.1)	96 (38.7)	152 (61.3)	Reference	
Positive	54 (17.9)	17 (31.5)	37 (68.5)	0.01 (−0.74, 0.76)	0.98
Asymptomatic malaria between birth and 5 years (*n*, %)					
Negative	268 (88.7)	103 (38.4)	165 (61.6)	Reference	
Positive	34 (11.3)	10 (29.4)	24 (70.6)	1.14 (0.27, 2.02)	0.01
Malaria episodes between birth and 5 years (*n*, %)					
None	135 (44.7)	56 (41.8)	79 (58.5)	Reference	
1	65 (21.5)	24 (36.9)	41 (63.1)	−0.38 (−1.12, 0.36)	
≥2	102 (33.8)	33 (32.3)	69 (67.7)	0.25 (−0.38, 0.89)	0.48 *
Haemoglobin levels at time of 25(OH)D measurement ^a^ (*n*, %) (mv = 16)					
Normal	172 (60.1)	62 (36.1)	110 (63.9)	Reference	
Anaemia	114 (39.9)	44 (38.6)	70 (61.4)	−0.17 (−0.74, 0.40)	0.56
Iron deficiency at time of 25(OH)D measurement ^b^ (*n*, %) (mv = 14)					
Normal	210 (72.9)	76 (36.2)	134 (63.8)	Reference	
Iron deficiency	78 (27.1)	31 (39.7)	47 (60.3)	0.03 (−0.60, 0.66)	0.93
CRP levels at time of 25(OH)D measurement ^c^ (*n*, %) (mv = 3)					
Normal	230 (76.9)	98 (42.6)	132 (57.4)	Reference	
Inflammation	69 (23.1)	15 (21.7)	54 (78.3)	0.79 (0.14, 1.46)	0.02
Randomized treatment of children with albendazole (ABZ) in EMaBS trial					
Placebo	143 (47.4)	53 (37.1)	90 (62.9)	Reference	
ABZ	159 (52.7)	60 (37.7)	99 (62.3)	0.13 (−0.41, 0.67)	0.63
Maternal age in years at enrolment to EMaBS (*n*, %)					
14–24	165 (54.6)	63 (38.2)	102 (61.8)	Reference	
25–34	110 (36.4)	40 (36.4)	70 (63.6)	−0.09 (−0.77, 0.59)	
35+	27 (8.9)	10 (37.0)	17 (63.0)	0.69 (−0.54, 1.91)	0.55 *
Maternal education at enrolment to EMaBS (*n*, %) (mv = 1)					
Primary/none	165 (54.8)	66 (40.0)	99 (60.0)	0.19 (−0.91, 1.29)	
Secondary	114 (37.9)	37 (32.5)	77 (67.5)	0.46 (−0.64, 1.56)	
Tertiary	22 (7.3)	9 (40.9)	13 (59.1)	Reference	0.77 *
Parity (*n*, %)					
1	54 (17.9)	27 (50.0)	27 (50.0)	Reference	
2–4	179 (59.3)	61 (34.1)	118 (65.9)	0.19 (−0.56, 0.94)	
5+	69 (22.9)	25 (36.2)	44 (63.8)	−0.09 (−1.23, 1.05)	0.97 *
Randomized treatment of mothers with ABZ during pregnancy in EMaBS trial (*n*, %)					
Placebo	144 (47.7)	51 (35.4)	93 (64.6)	Reference	
ABZ	158 (552.3)	62 (39.2)	96 (60.8)	−0.02 (−0.56, 0.52)	0.94
Randomized treatment of mothers with praziquantel during pregnancy in EMaBS trial (*n*, %)					
Placebo	167 (55.3)	65 (38.9)	102 (61.1)	Reference	
Praziquantel	135 (44.7)	48 (35.6)	87 (64.4)	0.47 (−0.07, 1.00)	0.09
Household socioeconomic status recorded at EMaBS enrolment ^d^ (*n*, %) (mv = 5)					
1 (lowest)	16 (5.4)	13 (81.3)	3(18.7)	−1.49 (−2.92, −0.07)	
2	16 (5.4)	6 (37.5)	10 (62.5)	−1.26 (−2.77, 0.24)	
3	82 (27.6)	22 (26.8)	60 (73.2)	0.33 (−0.71, 1.37)	
4	89 (29.9)	30 (33.7)	59 (66.3)	−0.14 (−1.18, 0.89)	
5	69 (23.2)	29 (42.0)	40 (58.0)	−0.33 (−1.39, 0.72)	
6 (highest)	25 (8.4)	11 (44.0)	14 (56.0)	Reference	0.25 *
Location recorded at EMaBS enrolment (*n*, %)					
Urban	118 (39.1)	49 (41.5)	69 (58.5)	Reference	
Peri-urban	79 (26.2)	29 (36.7)	50 (63.3)	0.36 (−0.33, 1.05)	
Rural	105 (34.8)	35 (33.3)	70 (66.7)	0.24 (−0.41, 0.89)	0.45 *

SD, standard deviation; mv, missing values. ^a^ Anaemia as haemoglobin <11 g/dL, adjusted for change in altitude (1000 m above sea level); ^b^ iron deficiency as ferritin levels <12 µg/L; and ^c^ inflammation as C-reactive protein levels >5 mg/L. ^d^ Household socioeconomic status was derived as a composite of building materials of the home, number of rooms and items owned. ††All multivariable models were adjusted for all the other participant characteristics (age at 25(OH)D measurement, sex, iron status, anaemia, inflammation, weight-for-height Z-scores, helminthic infections between birth and 5 years, randomized treatment of children and mothers with albendazole and praziquantel, household socioeconomic status, location, maternal age and education, parity and asymptomatic malaria parasitaemia. Sample size reduced from 302 to 295 children in the final multivariable analyses due to missing values in some of the adjusted variables such as anaemia and iron status. * *p* value for trend.

**Table 3 nutrients-12-01662-t003:** Multivariable linear regression results for associations between participant characteristics and outcomes.

Variables	Verbal and Non-Verbal IQ (*n* = 259) β (95% CI)	*p* Value	Executive Function (*n* = 259) β (95% CI)	*p* Value	Motor Function (*n* = 259) β (95% CI)	*p* Value
Sex						
Male	Reference		Reference		Reference	
Female	−0.19 (−0.50, 0.13)	0.25	0.42 (0.11, 0.72)	0.01	0.33 (0.05, 0.60)	0.02
HAZ						
Normal	Reference		Reference		Reference	
Stunted	−0.30 (−0.64, 0.04)	0.08	0.02 (−0.31, 0.35)	0.91	−0.05 (−0.34, 0.25)	0.75
Maternal education						
Primary/none	−0.46 (−1.08, 0.16)		−0.43 (−1.04, 0.17)		0.27 (−0.27, 0.81)	
Secondary	−0.21 (−0.84, 0.41)		−0.18 (−0.79, 0.43)		0.34 (−0.21, 0.88)	
Tertiary	Reference	0.07 *	Reference	0.07 *	Reference	0.66 *
Household socioeconomic status						
1 (Lowest)	−0.33 (−1.19, 0.53)		−0.04 (−0.88, 0.79)		0.36 (−0.38, 1.11)	
2	0.09 (−0.80, 0.99)		−0.04 (−0.91, 0.31)		0.17 (−0.61, 0.94)	
3	−0.06 (−0.68, 0.57)		−0.29 (−0.90, 0.32)		0.37 (−0.17, 0.92)	
4	0.06 (−0.56, 0.69)		−0.23 (−0.84, 0.38)		0.42 (−0.12, 0.97)	
5	0.44 (−0.19, 1.08)		0.10 (−0.52, 0.72)		0.39 (−0.16, 0.94)	
6 (Highest)	Reference	0.08 *	Reference	0.35 *	Reference	0.66 *
Location						
Urban	Reference		Reference		Reference	
Peri-urban	0.03 (−0.37, 0.43)		−0.37 (−0.76, 0.02)		−0.08 (−0.42, 0.27)	
Rural	−0.11 (−0.49, 0.27)	0.57 *	−0.20 (−0.57, 0.17)	0.27 *	0.12 (−0.21, 0.45)	0.50 *
Randomized treatment of children with albendazole (ABZ) in EMaBS trial						
ABZ	Reference		Reference		Reference	
Placebo	−0.07 (−0.39, 0.25)	0.69	−0.13 (−0.45, 0.18)	0.40	−0.31 (−0.59, −0.04)	0.03

CI, confidence interval; IQ, intelligence quotient; HAZ, height-for-age Z-scores. All models are adjusted for age at 25(OH)D measurement, 25(OH)D levels, anaemia, iron status, inflammation and asymptomatic malaria. Sample size reduced from 302 to 259 children in the final multivariable analyses due to missing values in some of the adjusted variables such as anaemia and iron status. * *p* value for trend.

**Table 4 nutrients-12-01662-t004:** Univariable and multivariable linear regression results for the association between 25(OH)D levels and cognitive and motor outcomes.

	Univariable Model β (95% CI) *n*= 302	*p* Value	Model 1 * β (95% CI) *n* = 272	*p* Value	Model 2 ** β (95% CI) *n* = 259	*p* Value
**Verbal and non-verbal IQ**						
25(OH)D levels (per 10 nmol/L)	−0.01 (−0.07, 0.06)	0.76	0.01 (−0.07, 0.08)	0.86	0.01(−0.07, 0.08)	0.82
25(OH)D levels >75 nmol/L	Reference		Reference		Reference	
25(OH)D levels 50–75 nmol/L	−0.08 (−0.40, 0.24)	0.63	0.02 (−0.31, 0.35)	0.89	0.04 (−0.30, 0.39)	0.80
25(OH)D levels ≤75 nmol/L	0.08 (−0.23, 0.39)	0.63	−0.01 (−0.33, 0.32)	0.96	−0.05 (−0.38, 0.29)	0.78
**Executive function**						
25(OH)D levels (per 10 nmol/L)	0.01 (−0.06, 0.07)	0.81	0.03 (−0.04, 0.10)	0.37	0.04 (−0.03, 0.12)	0.25
25(OH)D levels >75 nmol/L	Reference		Reference		Reference	
25(OH)D levels 50–75 nmol/L	−0.07 (−0.38, 0.23)	0.64	−0.06 (−0.38, 0.27)	0.73	−0.04 (−37, 0.29)	0.81
25(OH)D levels ≤75 nmol/L	0.08 (−0.22, 0.39)	0.58	0.04 (−0.27, 0.35)	0.80	0.01 (−0.32, 0.34)	0.94
**Motor function**						
25(OH)D levels (per 10 nmol/L)	0.02 (−0.04, 0.07)	0.54	0.02 (−0.04, 0.08)	0.55	0.02 (−0.04, 0.09)	0.52
25(OH)D levels >75 nmol/L	Reference		Reference		Reference	
25(OH)D levels 50–75 nmol/L	0.00 (−0.27, 0.27)	0.99	0.01 (−0.27, 0.29)	0.96	0.01 (−0.29, 0.30)	0.97
25(OH)D levels ≤75 nmol/L	−0.04 (−0.29, 0.22)	0.79	−0.06 (−0.33, 0.22)	0.68	−0.06 (−0.34, 0.23)	0.70

CI, confidence interval. * Model 1 was minimally adjusted for age at 25(OH)D measurement, sex, iron status and anaemia. ** Model 2 was adjusted for factors in model 1 plus stunting, inflammation, randomized treatment of children with albendazole, socioeconomic status, maternal education and asymptomatic malaria parasitaemia. Sample size reduced from 302 to 259 children in the final multivariable analyses due to missing values in some of the adjusted variables such as anaemia and iron status. All *p* values are from the Wald test.

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
