# Peer review of "Vitamin D Status Is Not Associated with Cognitive or Motor Function in Pre-School Ugandan Children"

_nutrients, 2020, doi:10.3390/nu12061662_

Round 1

Reviewer 1 Report

This manuscript is well written and I commend the authors for clarity.  There are no issues with the overall scientific merit or their methods.  I do have concerns with the small sample size but the effect sizes appear to override this point.  The vitamin D debate continues and I welcome the paper as a contribution to the literature despite the findings of no effect.

Author Response

Response to Reviewer 1 Comments

Point 1: This manuscript is well written and I commend the authors for clarity.  There are no issues with the overall scientific merit or their methods.  I do have concerns with the small sample size but the effect sizes appear to override this point.  The vitamin D debate continues and I welcome the paper as a contribution to the literature despite the findings of no effect.

Response 1: We thank the reviewer for the kind comments. We have now included a sample size calculation in the Methods section as follows:

Materials and Methods under Sample Size (lines 108-113): “In the larger EMaBS study, the cohort size of 2,500 was determined based on the primary immunological objectives. For the present sub-study, assuming a prevalence of 36.6% of 25(OH)D levels ≤75 nmol/L among Ugandan children [21] and a standard deviation of two in cognitive and motor scores [22], our sample size of 302 has 80% power at α=0.05 to detect a difference of 0.7 when comparing mean cognitive / motor scores between the 25(OH)D levels ≤75 nmol/L and >75 nmol/L groups.”

Reviewer 2 Report

This is an interesting and well-conducted study but some issues have to be considered.

  1. The authors define vitamin D insufficiency as “levels 50-75 nmol/L”. However, this is not a generally accepted definition and is not based on sound scientific evidence. Most commonly vitamin D insufficiency or deficiency is defined as below ≥50 nmol/l and sufficiency as above 50 nmol/l. This issue should be corrected throughout the manuscript. To present results using the cut-off of 75 nmol/l is ok but the definition is not correct. I would report this as e.g. “…105 (35.8%) had 25OHD concentrations between 50 and 75 nmol/L”
  2. This also needs justification why did you choose this cut-off of 75 nmol/l in the final models?
  3. Line 61: “Vitamin D deficiency is also associated with structural and functional alterations 61 in the developing brains of rodents [11-13]”. What kind of implications could these alterations have on (clinical) outcomes in rat or human?
  4. Line 85: “…parents or guardians provided written informed consent.” Were all parents literate?
  5. What is the basis that vitamin D could be one of the most interesting factor affecting cognitive abilities from all other environmental factors? Considering that these children most likely have sufficient sunlight induced vitamin D synthesis but other issues like infections and undernutrition may have much stronger effect on your outcomes?
  6. How did you consider the possibility that those children who are primarily already more active in motor and cognitive skills – they are also more outside in the sun…
  7. Were there intervention effects on these outcomes?
  8. socioeconomic status – how was this covariate created, e.g. did you count living conditions as housing/rural/urban?
  9. Line 161: “We empirically tested associations of potential confounders with 25(OH)D levels and cognitive and motor outcomes using univariable linear regression models”. What were these confounders you tested? You report covariates you found based on literature but not these.
  10. Line 182: there were quite a few “Error! Reference source not found” from this point forward.
  11. FigureS1: Great that you tried this and I feel this is important figure but it may too much in a single manuscript. It lacks quite a lot of important arrows and factors, e.g. vitamin D status can affect anthropometry, inflammation can affect diet, diet on cognitive and motor outcomes, prenatal state? etc. You could try to narrow down the question OR have a more general perspective and separate factors could be presented as examples OR add explanations and justifications OR do it in a more comprehensible way altogether OR remove it.
  12. You may say this in a more understandable way: “Insufficient 25(OH)D levels were less common among children with inflammation” E.g. Children with 25OHD above 75 nmol/l had higher CRP levels than those with 25OHD below 75 nmol/l(?) /or inflammation was more common…
  13. Was location inquired at study baseline? What about the situation at 25OHD measurement? Could residence or socioeconomic status have changed since baseline and affect results?
  14. Line 199: “Baseline characteristics were comparable between participants included in the analyses and those not included” Where are these results?
  15. Table 2 vs Table 3: Why did you chose different tests in these analyses? Table 2 is descriptive table with proportions but table 3 present results on multivariate linear regression. I would like to see also multivariate linear regression on the factors associating with 25OHD.
  16. Line 232: This Figure S1 as one result is confusing. Figure needs clear aim and description, and visible results which it is based.
  17. Line 235: Something missing?
  18. What is the added value of table S3 compared to table 3?
  19. What is the basis of showing certain covariates in table 2 and other in table S1? You should simplify your results or have at least a clearer structure on what to present in manuscript and what in supplementary files.
  20. Line 257: …”although vitamin D insufficiency was common (34.8%).” Please remove this and emphasize that the vitamin D status was sufficient in this population. -And this is a positive thing! and maybe expected at this age group at this geographical location?
  21. Line 267: Do you mean Table S4 -not S1? Table S4 is not needed in this manuscript.
  22. Discussion has to be clearly shorter and more concise. Paragraphs 2, 3 and 6 has repetition – all this information could be just one paragraph.
  23. Line 344-351: “Though considered the most reliable marker of vitamin D status, 25(OH)D has a half-life of three 344 weeks [60] and is likely to change with age, exposure to sunlight and diet.” Do you think this is a relevant limitation in your study? How all those limitations could have affected your results?
  24. Limitation is the large drop-out: 2507 recruited and in this secondary analysis only 300. This should be discussed.
  25. Line 353: “In conclusion, the results from this study provide no evidence of association between 25(OH)D 353 levels measured below five years of age and cognitive or motor outcomes in Ugandan children at five 354 years of age.” -I’m surprised. You show results, done accordingly, which are in line with previous studies so shouldn’t your conclusion be that your study adds to the evidence that vitamin D status has no role/association with cognitive or motor outcomes …at least in this population with sufficient vitamin D status.
  26. Line 357: “For instance, well-conducted randomized controlled trials of vitamin D supplementation would be valuable in clarifying the causal effect of vitamin D in early childhood for cognitive and motor development.” Why do you think this is needed if in general no association has been observed? Maybe clarify what you mean. Could some other studies be done before an intervention regarding these outcomes?
  27. Last sentence is too long. Also, I don’t quite understand why do you emphasize the need of “incorporating other domains of development”?

Author Response

We thank the reviewers for their helpful comments, which we have addressed point-by-point below:

Response to Reviewer 2 Comments

Point 1: This manuscript is well written and I commend the authors for clarity.  There are no issues with the overall scientific merit or their methods.  I do have concerns with the small sample size but the effect sizes appear to override this point.  The vitamin D debate continues and I welcome the paper as a contribution to the literature despite the findings of no effect.

Response 1: We thank the reviewer for the kind comments. We have now included a sample size calculation in the Methods section as follows:

Materials and Methods under Sample Size (lines 108-113): “In the larger EMaBS study, the cohort size of 2,500 was determined based on the primary immunological objectives. For the present sub-study, assuming a prevalence of 36.6% of 25(OH)D levels ≤75 nmol/L among Ugandan children [21] and a standard deviation of two in cognitive and motor scores [22], our sample size of 302 has 80% power at α=0.05 to detect a difference of 0.7 when comparing mean cognitive / motor scores between the 25(OH)D levels ≤75 nmol/L and >75 nmol/L groups.”

Response to Reviewer 2 Comments

Point 1: The authors define vitamin D insufficiency as “levels 50-75 nmol/L”. However, this is not a generally accepted definition and is not based on sound scientific evidence. Most commonly vitamin D insufficiency or deficiency is defined as below ≥50 nmol/l and sufficiency as above 50 nmol/l. This issue should be corrected throughout the manuscript. To present results using the cut-off of 75 nmol/l is ok but the definition is not correct. I would report this as e.g. “…105 (35.8%) had 25OHD concentrations between 50 and 75 nmol/L”

Response 1: Thank you. We have now changed the wording for the 25(OH)D categories throughout the manuscript as follows:

Abstract (lines 38-40): “Of 302 children, 8 (2.7%) had 25(OH)D levels <50 nmol/L, 105 (35.8%) had levels 50-75 nmol/L and 189 (62.6%) had levels >75 nmol/L.”

Materials and Methods, under Laboratory Procedures and Definitions (lines 118-121): “For this study, we classified 25(OH)D levels into three categories; levels <50 nmol/L, 50-75 nmol/L, and >75 nmol/L, according to the Endocrine Society Practice Guidelines [24]”.

Materials and Methods, under Statistical Analysis (lines 160-162): “We examined the characteristics of the participants using descriptive statistics including proportions and means with standard deviations and used ANOVA to compare mean scores between children with 25(OH)D levels <50 nmol/L, 50-75 nmol/L and >75 nmol/L”.

Results, under Description of Study Participants (lines 206-208): “Eight (2.7%) children had 25(OH)D levels <50 nmol/L, 105 (34.8%) had levels 50-75 nmol/L and 189 (62.6%) had levels >75 nmol/L.”

Results, under Associations between 25(OH)D and Cognitive and Motor Outcomes (lines 272-280): “Considering 25(OH)D as a categorical variable, we found no evidence of differences in the mean scores among children with 25(OH)D levels <50 nmol/L, 50-75 nmol/L and >75 nmol/L for IQ (P=0.89), EF (P=0.44) or motor function (P=0.44) outcomes. We found little evidence of association between IQ, EF or motor function and 25(OH)D levels ≤75 nmol/L compared to levels >75 nmol/L or for levels 50-75 nmol/L compared to levels >75 nmol/L after adjusting for potential confounders (Table 4).

Table 4: We have changed all the labels from vitamin D insufficient and vitamin D sufficient to 25(OH)D levels ≤75 nmol/L and 25(OH)D levels >75 nmol/L respectively and have additionally added analyses for 25(OH)D levels 50-75 nmol/L.

Discussion (lines 298-300): “In this study population, we found that few children had 25(OH)D levels <50 nmol/L (2.7%), and the majority of children (62.6%) had 25(OH)D levels >75 nmol/L.”

Discussion (lines 302-305): “This finding may be explained by the small proportion of children with 25(OH)D levels <50 nmol/L limiting the ability to detect an effect, especially if 25(OH)D levels <50 nmol/L rather than 25(OH)D levels ≤75 nmol/L drive impaired neurobehavioural outcomes [34].”

Discussion (lines 417-420): “We were unable to assess the association between 25(OH)D levels <50nmol/L and cognitive and motor outcomes as only 2.7% children had 25(OH)D levels <50 nmol/L in this study population.”

Point 2: This also needs justification why did you choose this cut-off of 75 nmol/l in the final models?

Response 2: Our cut-offs for 25(OH) were based on the Endocrine Society Practice Guidelines. We chose a cut-off of 75 nmol/l in the final models since few (only 8) children had 25(OH)D levels <50 nmol/L in our population, therefore we could not compare 25(OH)D categories <50 nmol versus >50 nmol/L in our models. We have highlighted this as follows:

Materials and Methods, under Laboratory Procedures and Definitions (lines 118-121): “For this study, we classified 25(OH)D levels into three categories; levels <50 nmol/L, 50-75 nmol/L, and >75 nmol/L according to the Endocrine Society Practice Guidelines [24]”.

Materials and Methods, under Statistical Analysis (lines 186-189): “We were unable to further investigate associations with 25(OH)D levels <50 nmol/L using multivariable analyses since few children (n=8) had 25(OH)D levels <50 nmol/L, so children with 25(OH)D levels <50 nmol/L were grouped with children that had levels 50-75 nmol/L. We also compared children with 25(OH)D levels 50-75 nmol/L with children that had levels >75 nmol/L.”

Results, under Associations between 25(OH)D and Cognitive and Motor Outcomes (lines 277-280): “ We found little evidence of association between IQ, EF or motor function and 25(OH)D levels ≤75 nmol/L compared to levels >75 nmol/L or for levels 50-75 nmol/L compared to levels >75 nmol/L after adjusting for potential confounders (Table 4).”

Table 4: We have included results for the associations between cognitive / motor outcomes with 25(OH)D levels 50-75 nmol/L compared to levels >75 nmol/L.

Point 3: Line 61: “Vitamin D deficiency is also associated with structural and functional alterations 61 in the developing brains of rodents [11-13]”. What kind of implications could these alterations have on (clinical) outcomes in rat or human?

Response 3: Thank you. We have now added information on the possible implications of these alterations as follows:

Introduction (lines 61-63): “Vitamin D deficiency is also associated with structural and functional alterations in the developing brains of rodents, which are associated with impaired behaviour, learning, motor and cognitive processes [11-13].”

Point 4: Line 85: “…parents or guardians provided written informed consent.” Were all parents literate?

Response 4: We have now included the following statement:

Materials and methods under Ethical Approval (lines 86-88): “Parents or guardians provided informed consent in writing, or with a thumbprint if not literate with a signature from a literate witness.”

Point 5: What is the basis that vitamin D could be one of the most interesting factor affecting cognitive abilities from all other environmental factors? Considering that these children most likely have sufficient sunlight induced vitamin D synthesis but other issues like infections and undernutrition may have much stronger effect on your outcomes?

Response 5: We agree that many other factors such as poverty, undernutrition and infection are likely to affect child development. We therefore also assessed for the effects of socioeconomic status, undernutrition and common childhood infections such as malaria and helminthic infections on child development and we note that there may also be residual confounding by other unmeasured factors. We hypothesised that vitamin D may be important for cognitive and motor function based on consistent evidence from animal studies suggesting an important role for vitamin D in brain development (Introduction, lines 58-64). Moreover, there is sparse literature on the role of vitamin D on neurodevelopment in African children and two other studies found associations between 25(OH)D levels and cognition, school performance and behaviour in African children (Introduction, lines 70-76). A recent systematic review and meta-analysis also showed that inadequate 25(OH)D levels may be common in African populations (Introduction, lines 54-57). However, we found that the children in this study had generally adequate 25(OH)D levels. We have made the following changes:

Discussion (lines 425-431): “Further, the possibility of other unmeasured variables that may affect 25(OH)D levels cannot be ruled out. These include habits that might influence exposure to sunlight, for example, outdoor physical activity and clothing and dietary vitamin D intake. However, diet may not be an important risk factor since vitamin D fortification and supplementation is not routine in Uganda. There may have also been other unmeasured factors that might have influenced cognitive and motor outcomes, such as pre-schooling, inadequately stimulating home environment or maternal IQ.”

Point 6: How did you consider the possibility that those children who are primarily already more active in motor and cognitive skills – they are also more outside in the sun…

Response 6: We agree that there is the possibility of reverse causality, whereby children with better motor or cognitive function are likely to spend more time outdoors for example playing and consequently have higher 25(OH)D levels compared to children with poor motor or cognitive function. However, in this study we did not find any evidence of association between higher 25(OH)D levels and improved cognitive / motor function.

Point 7: Were there intervention effects on these outcomes?

Response 7: None of the participants in our study received vitamin D supplementation. We observed marginal effects of intervention with albendazole on cognitive and motor outcomes in the original trial as indicated in the following statements:

Results, under Description of Study Participants (lines 199-200): “None of the included children or their mothers received vitamin D supplementation.”

Results, under Cognitive and Motor Outcomes at Five Years and Associations with Participant Characteristics (lines 251-254): “Children who were randomly allocated placebo in the original EMaBS trial had lower motor scores compared to those who received albendazole but little difference was observed in IQ and EF scores (Table 3).”

Discussion (lines 387-391): “Furthermore, randomized treatment of children with albendazole in the EMaBS trial was marginally associated with higher motor scores compared to placebo treatment. This may be a chance finding since in the larger EMaBS birth cohort no associations were observed between either child or maternal treatment with albendazole and cognitive or motor scores [64, 65].”

Point 8: Socioeconomic status – how was this covariate created, e.g. did you count living conditions as housing/rural/urban?

Response 8: We have included this information as follows:

Table 2 legend (lines 237-238): “Household socioeconomic status was derived as a composite of building materials of the home, number of rooms and items owned.”

Materials and Methods, under Participants and Study Design (lines 103-106):

“Household socioeconomic status was derived as a composite of building materials of the home, number of rooms and items owned. A total asset score was calculated and categorized into 6 ordered categorical levels with 1 indicating lowest and 6 indicating highest household socioeconomic status.”

Point 9: Line 161: “We empirically tested associations of potential confounders with 25(OH)D levels and cognitive and motor outcomes using univariable linear regression models”. What were these confounders you tested? You report covariates you found based on literature but not these.

Response 9: We tested all the potential confounders identified from literature as listed plus the other participant characteristics. We have now clarified the statements as follows:

Materials and Methods, under Statistical Analysis (line 168-178): “We identified potential confounders from literature including age at 25(OH)D measurement, sex, iron status and anaemia (haemoglobin levels <11 g/dl), inflammation (CRP levels >5 mg/L), height-for-age Z-scores, randomized treatment of children with albendazole, socioeconomic status, maternal education and asymptomatic malaria parasitaemia. We empirically tested associations of all the potential confounders identified from literature in addition to weight-for-age and weight-for-height Z-scores, location, maternal age, parity and randomized maternal treatment with albendazole and praziquantel with 25(OH)D levels and cognitive and motor outcomes using multivariable linear regression models.”

Point 10: Line 182: there were quite a few “Error! Reference source not found” from this point forward.

Response 10: Thank you for noting this. We have now corrected the error phrases into the respective Tables referred to, that is Table 2 (lines 201 and 214) and Table 3 (lines 251 and 254).

Point 11: FigureS1: Great that you tried this and I feel this is important figure but it may too much in a single manuscript. It lacks quite a lot of important arrows and factors, e.g. vitamin D status can affect anthropometry, inflammation can affect diet, diet on cognitive and motor outcomes, prenatal state? etc. You could try to narrow down the question OR have a more general perspective and separate factors could be presented as examples OR add explanations and justifications OR do it in a more comprehensible way altogether OR remove it.

Response 11: Thank you for these suggestions. We created Figure S1 to visually explore the associations between vitamin D, covariates and cognitive/motor outcomes as a step towards identifying potential confounders to adjust for in the multivariable analyses. We agree that Figure S1 may be complicated for our main research question (associations between 25(OH)D levels and cognitive and motor outcomes) and have now removed the figure and the following related text from the manuscript:

Materials and Methods, under Statistical analysis (lines 172-174): “and generated a directed acyclic graph (DAG) [31] (Supplementary Figure S1) to explore the hypothesized associations between vitamin D, outcomes and covariates.”

Results, under Associations between 25(OH)D and Cognitive and Motor Outcomes (lines 266-268): “We generated a directed acyclic graph that further confirmed hypothesized associations between 25(OH)D levels and cognitive and motor outcomes and identified potential confounders (Supplementary Figure S1).”

Point 12: You may say this in a more understandable way: “Insufficient 25(OH)D levels were less common among children with inflammation” E.g. Children with 25OHD above 75 nmol/l had higher CRP levels than those with 25OHD below 75 nmol/l(?) /or inflammation was more common…

Response 12: We have now changed the statement to make it more understandable as follows:

Results, under Associations Between 25(OH)D and Participant characteristics (lines 228-229): “Inflammation was associated with higher mean 25(OH)D levels and 25(OH)D levels >75 nmol/L were more common among those with inflammation than for those with lower CRP levels (Table 2).”

Point 13: Was location inquired at study baseline? What about the situation at 25OHD measurement? Could residence or socioeconomic status have changed since baseline and affect results?

Response 13: Information on location was collected at enrolment but not at measurement of 25(OH)D levels. We have now included location in the data variables collected at enrolment and indicated that residence or socioeconomic status could have changed for some participants since baseline which may have affected our results, as follows:

Materials and Methods, under Participants and Study Design (line 103): “Information on maternal age, socioeconomic status, location, education, parity and medical conditions was collected at enrolment.”

Discussion (431-434): “Additionally, other variables recoded at enrolment such as location and socioeconomic status of some participants may have changed over the study period which may possibly result in residual confounding for the associations between 25(OH)D and cognitive and motor outcomes.”

Point 14: Line 199: “Baseline characteristics were comparable between participants included in the analyses and those not included” Where are these results?

Response 14: We have now added Supplementary Table S1 in the Supplementary Material showing a comparison of baseline characteristics between children who were included in this analysis and children who were excluded due to loss to follow up or death or missing 25(OH)D and / or cognitive and motor outcome data.

Point 15: Table 2 vs Table 3: Why did you chose different tests in these analyses? Table 2 is descriptive table with proportions but table 3 present results on multivariate linear regression. I would like to see also multivariate linear regression on the factors associating with 25OHD.

Response 15: We have now added results on multivariable associations between 25(OH)D and participant characteristics in Table 2 and added the following information in the Results and Discussion:

Results, under Associations Between 25(OH)D and Participant Characteristics (lines 226-229): “In the multivariable analyses we observed evidence of a decreasing trend of 25(OH)D levels with increasing child age and asymptomatic malaria was associated with higher 25(OH)D levels (Table 2). Inflammation was associated with higher mean 25(OH)D levels and 25(OH)D levels >75 nmol/L were more common among those with inflammation than for those with lower CRP levels (Table 2).”

Discussion (lines 376-382): “We observed a decreasing trend of 25(OH)D levels with increasing child age consistent with previous evidence [56]. Younger children may acquire 25(OH)D from their mothers through placental transfer of 25(OH)D in utero or breastfeeding during infancy. Asymptomatic malaria parasitaemia was associated with higher 25(OH)D levels. Evidence on the relationship between malaria and 25(OH)D levels is limited and inconclusive [57, 58].  High 25(OH)D levels were associated with increased inflammation contrary to other studies that have shown possible anti-inflammatory properties of vitamin D [59].”

Point 16: Line 232: This Figure S1 as one result is confusing. Figure needs clear aim and description, and visible results which it is based.

Response 16: We created Figure S1 to visually explore the associations between vitamin D, covariates and cognitive/motor outcomes as a step towards identifying potential confounders to adjust for in the multivariable analyses. We agree that Figure S1 may be complicated for our main message (associations between 25(OH)D levels and cognitive and motor outcomes) and have now removed the figure and the following related text from the manuscript:

Results, under Associations between 25(OH)D and Cognitive and Motor Outcomes (lines 266-268): “We generated a directed acyclic graph that further confirmed hypothesized associations between 25(OH)D levels and cognitive and motor outcomes and identified potential confounders (Supplementary Figure S1).”

Point 17: Line 235: Something missing?

Response 17: We have added the missing information (line 269): “Table 4”

Point 18: What is the added value of table S3 compared to table 3?

Response 18: Table S3 shows the univariable associations between all participants' characteristics with the outcomes while Table 3 includes the multivariable associations with only factors that were found to be associated with the outcomes.

Point 19: What is the basis of showing certain covariates in table 2 and other in table S1? You should simplify your results or have at least a clearer structure on what to present in manuscript and what in supplementary files.

Response 19: We have now combined Table 2 and Table S1 into one Table (Table 2). We have also included the following statement in results:

Results, under Description of Study Participants (lines 200-201): “Characteristics of the study participants are presented in Table 2.”

Point 20: Line 257: …”although vitamin D insufficiency was common (34.8%).” Please remove this and emphasize that the vitamin D status was sufficient in this population. -And this is a positive thing! and maybe expected at this age group at this geographical location?

Response 20: We have now rephrased the statement as follows:

Discussion (lines 298-300): “In this study population, we found that few children had 25(OH)D levels <50 nmol/L (2.7%), and the majority of children (62.6%) had 25(OH)D levels >75 nmol/L.”

Point 21: Line 267: Do you mean Table S4 -not S1? Table S4 is not needed in this manuscript.

Response 21: We have now removed Supplementary Table S4 from the manuscript.

Point 22: Discussion has to be clearly shorter and more concise. Paragraphs 2, 3 and 6 has repetition – all this information could be just one paragraph.

Response 22: We have now combined the 3 paragraphs as follows:

Discussion (lines 309-330): “Our findings of no association between vitamin D status and cognitive or motor function, agree with nine previous observational studies that similarly reported no evidence of association in children [35-43]. On the contrary, a small dose-response randomized controlled trial (n=55) of vitamin D supplementation in Canadian infants reported that a lower dose of 400 international units (IU) of vitamin D was more beneficial for gross motor development than higher doses of 800 IU or 1200 IU [16]. This finding may indicate a threshold effect for vitamin D but may be limited by the small sample size and lack of a control group that did not receive vitamin D supplements. Additionally, a cross-sectional study of one-year-old Iranian children (n=186) reported a positive association between high 25(OH)D levels and improved motor function [44]. To our knowledge, ours is the first prospective study to investigate the relationship between vitamin D status and cognitive and motor development in pre-school children in sub-Saharan Africa. To date, only three observational studies have evaluated the effect of child or maternal vitamin D status on neurobehavioural outcomes in African children. One prospective study (n=254) observed an association between increasing 25(OH)D levels and improved socioemotional adjustment in Ugandan school-aged children who were HIV negative and those perinatally exposed to HIV but uninfected and unexposed to early antiretroviral therapy (ART) [18]. They also observed an association between increasing 25(OH)D levels and poor socioemotional adjustment in HIV-infected children and in HIV-uninfected children exposed to early ART, which may be attributed to dysregulation of 25(OH)D metabolism by ART. Another small study (n=45) reported that high 25(OH)D levels were positively associated with improved cognition in Egyptian school-aged children [19]. Additionally, an observational study of 202 mother-child pairs in Seychelles reported no association between maternal 25(OH)D levels during pregnancy and neurobehavioural outcomes at five years [45].”

Point 23: Line 344-351: “Though considered the most reliable marker of vitamin D status, 25(OH)D has a half-life of three 344 weeks [60] and is likely to change with age, exposure to sunlight and diet.” Do you think this is a relevant limitation in your study? How all those limitations could have affected your results?

Response 23: We note that this limitation may be relevant in this study as misclassification of vitamin D status, which was unknown at the time of outcome assessment may have resulted in an underestimation of the effect size (non-differential misclassification). We have changed the statement as follows:

Discussion (lines 420-423): “Additionally, plasma 25(OH)D levels were measured at one time-point prior to developmental assessment and may not reflect the vitamin D status of the children throughout early childhood or at the time of cognitive and motor assessment, and could lead to underestimation of the association between 25(OH)D and cognitive or motor outcomes.”

Point 24: Limitation is the large drop-out: 2507 recruited and in this secondary analysis only 300. This should be discussed.

Response 24: We have now added this information as follows:

Discussion (lines 412-417): “We included about 13% of participants in the original EMaBS birth cohort in our analyses as 25(OH)D levels and cognitive and motor outcomes were assessed in smaller subsets of participants and over a quarter of the original participants were lost to follow up. However, our study was adequately powered to detect the reported effect sizes, and selection bias was unlikely as most baseline characteristics were comparable between children who were included in these analyses and those who were not included.”

Point 25: Line 353: “In conclusion, the results from this study provide no evidence of association between 25(OH)D 353 levels measured below five years of age and cognitive or motor outcomes in Ugandan children at five 354 years of age.” -I’m surprised. You show results, done accordingly, which are in line with previous studies so shouldn’t your conclusion be that your study adds to the evidence that vitamin D status has no role/association with cognitive

Response 25: We have now edited the statement as follows:

Conclusions (lines 437-438): “In conclusion, our findings add to the evidence from similar studies that report no association between vitamin D status and cognitive and motor outcomes in children.”

Point 26: Line 357: “For instance, well-conducted randomized controlled trials of vitamin D supplementation would be valuable in clarifying the causal effect of vitamin D in early childhood for cognitive and motor development.” Why do you think this is needed if in general no association has been observed? Maybe clarify what you mean. Could some other studies be done before an intervention regarding these outcomes?

Response 26: Thank you for this comment. We agree that further longitudinal studies might be useful in estimating the associations between 25(OH)D levels and cognitive / motor outcomes particularly in populations with high prevalence of vitamin D deficiency. However, we also note that evidence from similar observational studies is inconclusive and may be subject to residual confounding. Currently there are only 2 small randomized controlled trials investigating the effect of vitamin D supplementation on cognitive/ motor outcomes and hence larger well-conducted vitamin D supplementation trials would be useful in establishing these effects in children. We have edited the information as follows:

Conclusions (lines 442-447): “Further longitudinal studies in vitamin D deficient populations and well-conducted randomized controlled trials of vitamin D supplementation would be valuable in clarifying the causal effect of vitamin D status in early childhood for cognitive and motor development.”

Point 27: Last sentence is too long. Also, I don’t quite understand why do you emphasize the need of “incorporating other domains of development”?

Response 27: Thank you for this comment, we have now removed the last sentence (lines 447-450).

Round 2

Reviewer 2 Report

Thank you, alterations and explanations are adequate and this is an important study. If you wish you can use the definition deficiency for <50 nmol/l. I meant in my earlier comment the definition of 75 nmol/l which has been now corrected. 
